# Improved Strong Tracking Cubature Kalman Filter for UWB Positioning

**DOI:** 10.3390/s23177463

**Published:** 2023-08-28

**Authors:** Yuxiang Pu, Xiaolong Li, Yunqing Liu, Yanbo Wang, Suhang Wu, Tianshuai Qu, Jingyi Xi

**Affiliations:** 1Institute of Electronic Information Engineering, Changchun University of Science and Technology, Changchun 130022, China; pxy8051@163.com (Y.P.); mzliuyunqing@163.com (Y.L.); wang1516171819@163.com (Y.W.); wshwzl19970717@163.com (S.W.); 2022100839@mails.cust.edu.cn (T.Q.); m18844419310@163.com (J.X.); 2Jilin Provincial Science and Technology Innovation Center of Intelligent Perception and Information Processing, Changchun 130022, China

**Keywords:** UWB positioning, NLOS signal, cubature Kalman filter (CKF), strong tracking filtering, fading factor

## Abstract

For the problems of Non-Line-of-Sight (NLOS) observation errors and inaccurate predictive dynamics model in wireless ultra-wideband (UWB) positioning systems, an improved strong tracking cubature Kalman filter (ISTCKF) positioning algorithm is proposed in this paper. The main idea of the algorithm is as follows. First, the observations are reconstructed based on the weighted positioning results obtained from the predictive dynamics model and the least squares algorithm. Second, the difference in statistical properties between the observation noise and the NLOS errors is utilized to identify the NLOS observations by the corresponding judgment statistics obtained from the operation between the original observations and the reconstructed observations. The main positioning error of the UWB positioning system at the current moment is then judged by the NLOS identification results, and the corresponding fading factors are calculated according to the judgment results. Finally, the corresponding ISTCKF is constructed based on the fading factors to mitigate the main positioning error and obtain accurate positioning result in the UWB positioning system. In this paper, the reconstructed observations mitigate the observation noise in the original observation, and then the ISTCKF mitigates the main errors in the UWB positioning system. The experimental results show that the ISTCKF algorithm reduces the positioning error by 55.2%, 32.3% and 28.9% compared with STCKF, ACKF and RSTCKF, respectively. The proposed ISTCKF algorithm significantly improves the positioning accuracy and stability of the UWB system.

## 1. Introduction

With the rapid development of the Internet of Things and wireless technology, target positioning has become a popular research topic in recent years [1,2,3,4,5,6]. Wireless radio frequency positioning technologies such as Bluetooth [7,8,9], UWB [10,11,12], and Wi-Fi [13,14,15] are developing rapidly and have a great broad market application prospect in the society. Among them, UWB technology has received more and more attention for its advantages of low power consumption, high transmission rate, and strong penetration. At present, the ranging accuracy of UWB can reach the decimeter level under Line-of-Sight (LOS) conditions [16,17].

State estimation of nonlinear systems has a wide range of applications in the engineering field. Driven by this, a large number of filtering algorithms have been proposed. The algorithmic ideas are mainly divided into two categories: linearization methods and sampling methods. The result of the linearization method is the extended Kalman filter (EKF) [18]. The results of the linearization method include the unscented Kalman filter (UKF) [19], particle filter (PF) [20] and cubature Kalman filter (CKF) [21]. In particular, CKF has been shown to outperform the EKF, UKF, and PF [21].

In practical positioning scenarios, obstacles such as human body, concrete pillars, and trees in the signal transmission path can cause reflection, refraction, and diffraction of UWB wireless signals, which in turn will generate non-line-of-sight (NLOS) errors. The NLOS errors are one of the main errors affecting filtered positioning accuracy in UWB positioning [22]. In practical UWB filtering positioning, NLOS errors lead to low trustworthiness of the observations, which in turn causes a decrease in the filtering effectiveness and even the filter divergence [23]. Therefore, the processing of NLOS errors has been a hot research topic in the field of UWB positioning. For the processing of NLOS observations, firstly, the identification of NLOS observations is usually performed. Second, new observations are obtained by discarding the NLOS observations or mitigating the NLOS errors in the NLOS observations. Finally, the new observations are used to obtain more accurate localization results. The identification methods on the NLOS observations are broadly classified into three categories: based on statistical features [24,25,26], based on geometric constraints [27,28,29] and based on machine learning [30,31,32,33]. In the literature [25], Wylie performed long time observation statistics on observation measurements and found that the standard deviation values of observation measurement samples were much larger in the NLOS environment than in the LOS environment. Therefore, NLOS observations are identified by the standard deviation of the observation noise, but the method requires multiple measurements for identification. Therefore this method cannot meet the requirements of real-time identification of moving tags in practical application scenarios. In [26], Xie uses a soft judgment algorithm to classify UWB signals, where the identification of NLOS signals is based on the channel impulse response (CIR) characteristics of UWB signals. However, channel characteristics are affected by channel conditions, transmission distance, transmission power, etc. Luo proposed a geometrically constrained two-step LOS signal identification method based on the deviation of the common chord intersection from the mobile station position [29]. However, this method requires the computation of multiple positioning combinations and a more in-depth analysis of each positioning combination for the identification. Therefore, the identification algorithm for the geometrically constrained class is usually computationally intensive and has high computational complexity. Wei proposed a multi-input learning (MIL) neural network model based on channel impulse response (CIR) and time-frequency diagram of CIR (TFDOCIR) to identify NLOS signals in UWB positioning systems [32]. In [33], Bi proposes a Wi-Fi indoor positioning algorithm based on support vector regression (SVR) optimized by particle swarm optimization (PSO), termed PSOSVRPos. SVR algorithm devotes itself to solving localization as a regression problem by building the mapping between signal features and spatial co-ordinates in high dimensional space. The PSO algorithm concentrates on the global-optimal parameter estimation of the SVR model. However, This type of algorithm requires a large amount of training data and a high exchange power to analyze the channel statistical properties of the signal. Therefore, for real-time online applications in unknown environments, these methods have poor real-time performance. 

In most cases, NLOS observations are considered as outliers in UWB positioning algorithms and are therefore discarded. However, the identification and discard methods limit the effectiveness of UWB positioning systems in harsh and complex environments. If too many NLOS observations are discarded, usable observations are lost, which in turn reduces the accuracy of the UWB positioning system and even renders some UWB positioning algorithms inapplicable. Meanwhile, in the process of tag positioning, the randomness of tag motion and the uncertainty of its surrounding environment lead to variability in the predicted motion model and the unknown statistical properties of the observation noise. Therefore, when the associated errors cannot be well mitigated, the Kalman filter positioning algorithm will present large positioning errors and even cause filter divergence. Therefore, adaptive filtering, fading filtering, and robust filtering are often introduced into the Kalman filtering algorithm to mitigate the correlation errors and improve the accuracy of the positioning algorithm. For example, Gao adaptively adjusted and updated the prior information through the equivalent weighting matrix and adaptive factors to counteract the interference of system model errors on system state estimation, thus improving the state parameter estimation accuracy [34]. Zhao proposed a robust adaptive CKF (RACKF) to deal with the problem of system model inaccuracy and noise statistics by introducing adaptive factors and robust estimation theory [35]. Luo introduced a multi-fading factor into the covariance matrix of time update and measurement update, and adjusted the filter gain online in real time to reduce the degradation of filtering accuracy due to model mismatch [36]. However, the above algorithms are contradictory for the mitigation of the predictive dynamics model error and for the mitigation of the observation noise. This is because the predictive dynamics model error is mitigated by the observation information and the observation noise is mitigated by the predictive dynamics model. Therefore, it is necessary to identify the main source of the current UWB positioning errors and then adopt corresponding strategies to mitigate the errors.

In order to identify the main errors of UWB positioning and select a suitable strategy to mitigate the errors, an improved strong tracking CKF positioning algorithm is proposed in this paper. The contributions of this paper can be summarized as follows:The identification algorithm proposed in this paper solves the problem of insufficient real-time capability and applicability of some traditional identification algorithms in practical application scenarios. The identification algorithm in this paper does not require a priori information or multiple measurements to perform NLOS identification for each observation. The identification algorithm performs NLOS identification based on the reconstructed observation and the statistical feature difference between NLOS noise and LOS noise.The main error of the positioning system at the current moment can be distinguished by the results of the identification algorithm and the calculation of the conventional fading factor. The algorithm in this paper judges the main error of the localization system at the current moment, and then adopts the corresponding improved fading factor to enable the improved strong tracking cubature Kalman filtering algorithm to mitigate the main error of the localization system.Regarding the application of the algorithm, we designed simulation and field experiments to verify the stability and effectiveness of the proposed algorithm.

The rest of the paper is organized as follows. Section 2 describes the observation model and the CKF model. Section 3 details the ISTCKF algorithm. Section 4 describes the scenarios of the simulation and real experiments and their positioning results, and Section 5 and Section 6 present the discussion and conclusions of the paper.

## 2. Basic Theory

### 2.1. UWB Receive Measurement Model

A two-dimensional (2-D) positioning scenario is considered to model the UWB receive measurement. In the hybrid LOS/NLOS environment, the measurement model between the tag and the ith anchor node is:(1)$${\hat{d}}_{i}=\{\begin{array}{c} d_{i}+V_{i}=\sqrt{{\left(x-x_{i}\right)}^{2}+{\left(y-y_{i}\right)}^{2}}+V_{i},\;\;\;\;\;\mathrm{LOS}\;\mathrm{Propagation}\;\mathrm{state}\\ d_{i}+V_{i}+b_{{}_{i}}^{nlos}=\sqrt{{\left(x-x_{i}\right)}^{2}+{\left(y-y_{i}\right)}^{2}}+V_{i}+b_{{}_{i}}^{nlos},\;\mathrm{NLOS}\;\mathrm{Propagation}\;\mathrm{state} \end{array}$$
where $i=1,2,\ldots ,n_{y}$, and $n_{y}$ is the number of anchor nodes in the UWB positioning system; (x,y) and $\left(x_{i},y_{i}\right)$ are the coordinates of the tag and the *i*th anchor node; $d_{i}$ denotes the true distance between the tag and the *i*th anchor node; $V_{i}$ is the LOS observation noise which is modeled as Gaussian white noise with a mean of 0 and a variance of $\sigma_{los}^{2}$; $b_{i}^{nlos}$ is the NLOS noise in the NLOS environment. In the literature, the NLOS error is assumed to be positive and follows various distributions, e.g., uniform, Gaussian, and exponential distributions [37]. In this paper, we do not assume any distribution, i.e., our method can be applied under the assumption of any distribution. The only assumption here is that the NLOS error is much larger than the measurement noise. It is worth noting that this assumption is made in most existing works.

Thus, the observation noise $\zeta_{i}$ determined by the LOS and NLOS states can be express as:(2)$$\zeta_{i}=\{\begin{array}{c} V_{i}\;,\;\;\;\mathrm{LOS}\;\mathrm{Propagation}\;\mathrm{state}\\ V_{i}+b_{i}^{nlos}\;,\;\mathrm{NLOS}\;\mathrm{Propagation}\;\mathrm{state} \end{array}$$

On the other hand, considering the NLOS observations in the actual scenario, the NLOS errors can be fitted according to the NLOS observations obtained in the actual scenario test, and the simulated NLOS observations can be obtained more in line with the actual scenario. Figure 1 shows the NLOS error caused by pedestrians when the distance between the tag and the anchor node is fixed in a real-world scenario. The maximum NLOS error caused by the human body is about 120 cm, and the actual measured observation shows that there is a certain continuity of NLOS errors in the actual application scenario. Therefore, the proposed algorithm considers the NLOS errors with certain continuity.

### 2.2. Standard Cubature Kalman Filtering Algorithm

Considering the nonlinear discrete additive noise in the conventional case, the positioning system is modeled as:(3)$$\{\begin{array}{c} X_{k}=FX_{k-1}+w_{k}\\ Y_{k}=h(X_{k})+v_{k} \end{array}$$
where F and h(·) are the state transfer matrix and the transformation function of the state vector to the observation vector. $X_{k}\in R^{n_{x}}$ and $Y_{k}={\left[d_{1},d_{2},\ldots ,d_{n_{y}}\right]}^{T}\in R^{n_{y}}$ are the state vector and the observation vector, respectively. $w_{k}$ and $v_{k}$ are, respectively, predicted kinetic model error and observation noise, which are independent of each other and have a covariance matrix of $Q_{k}$ and $R_{k}$.

Considering CKF positioning problem in the two-dimensional case, the state vector at moment *k* is:$$X_{k}={\left[x_{k},y_{k},v_{xk},v_{yk}\right]}^{T}$$
where $x_{k}$ and $y_{k}$ indicate the position of the tag in the x- and y-directions, respectively; $v_{xk}$ and $v_{yk}$ denote the speed of motion of the tag in the x-direction and y-direction. 

#### 2.2.1. Prediction Update

For the prediction update, the filter state vector and the filter error covariance of the system at the moment *k* − *1* are $X_{k-1|k-1}$ and $P_{k-1|k-1}$. Then the CKF algorithm flow at moment *k* is as follows.

Decompose the filtering error covariance $P_{k-1|k-1}$ at moment *k* − *1* as follows:(4)$$P_{k-1|k-1}=S_{k-1|k-1}{\left(S_{k-1|k-1}\right)}^{T}$$
where $S_{k-1|k-1}$ is the result obtained by performing the Cholesky decomposition of $P_{k-1|k-1}$.

The cubature points are calculated as:(5)$$X_{i,k-1|k-1}=X_{k-1|k-1}+S_{k-1|k-1}\zeta_{i},i=1,2,\ldots ,m$$

The calculation of the cubature point through the nonlinear state transfer function is:(6)$$Xf_{i,k|k-1}=FX_{i,k-1|k-1},i=1,2,\ldots ,m$$
where $X_{i,k-1|k-1}$ and $Xf_{i,k|k-1}$ are the filtered state vector cubature points at moment *k* − *1* and the predicted state vector cubature points at moment *k*. *m* is the number of cubature points, according to the third-order cubature principle, there are $m=2n_{x}$. $\xi_{i}=\sqrt{m/2}{\left[1\right]}_{i}$ is the basic cubature point set, and ${\left[1\right]}_{i}$ is the *i*th column of matrix [1]. The set of points [1] can be expressed as: $$\left[1\right]=\left\{\begin{array}{cccc} \left(\begin{array}{c} 1\\ 0\\ \vdots \\ 0 \end{array}\right) & \left(\begin{array}{c} 0\\ 1\\ \vdots \\ 0 \end{array}\right) & ,\ldots , & \left(\begin{array}{c} 0\\ 0\\ \vdots \\ 1 \end{array}\right)\begin{array}{cccc} \left(\begin{array}{c} -1\\ 0\\ \vdots \\ 0 \end{array}\right) & \left(\begin{array}{c} 0\\ -1\\ \vdots \\ 0 \end{array}\right) & ,\ldots , & \left(\begin{array}{c} 0\\ 0\\ \vdots \\ -1 \end{array}\right) \end{array} \end{array}\right\}$$

The predicted state vector and the predicted state vector error covariance matrix are calculated as:(7)$$X_{k|k-1}=\frac{1}{m}\sum_{i=1}^{m}Xf_{i,k|k-1}$$
(8)$$P_{k|k-1}=\frac{1}{m}\sum_{i=1}^{m}Xf_{i,k|k-1}{\left(Xf_{i,k|k-1}\right)}^{T}-X_{k|k-1}{\left(X_{k|k-1}\right)}^{T}+Q_{k}$$

#### 2.2.2. Measurements Update

Cholesky decomposition of the predicted state vector error covariance matrix with:(9)$$P_{k|k-1}=S_{k|k-1}{\left(S_{k|k-1}\right)}^{T}$$

The corresponding cubature points obtained by cubature transformation of $X_{k|k-1}$ are:(10)$$X_{i,k|k-1}=X_{k|k-1}+S_{k|k-1}\xi_{i},i=1,2,\ldots ,m$$

The cubature points, calculated by h(·), are:(11)$$Y_{i,k|k-1}=h(X_{i,k|k-1}),i=1,2,\ldots ,m$$
where h(·) is the transition function between observation and state.

Calculate the predicted observation:(12)$$y_{k|k-1}=\frac{1}{m}\sum_{i=1}^{m}Y_{i,k|k-1}$$

Calculate the innovation and the innovation covariance matrix, respectively, as:(13)$$e_{k}=Y_{k}-y_{k|k-1}$$
(14)$$P_{yy,k|k-1}=\frac{1}{m}\sum_{i=1}^{m}Y_{i,k|k-1}Y_{{}_{i,k|k-1}}^{T}-y_{k|k-1}y_{{}_{k|k-1}}^{T}+R_{k}$$
where $y_{k}$ is the observation measurement at moment *k*.

The mutual covariance matrix between the predicted state vector and the predicted observation vector is calculated as:(15)$$P_{xy,k|k-1}=\frac{1}{m}\sum_{i=1}^{m}X_{i,k|k-1}Y_{i,k|k-1}^{T}-X_{k|k-1}y_{k|k-1}^{T}$$

Calculate the Kalman gain $K_{k}$, the state vector estimate $X_{k|k}$, and the filtering error covariance $X_{k|k}$ at moment *k*, denoted as:(16)$$K_{k}=P_{xy,k|k-1}P_{yy,k|k-1}^{-1}$$
(17)$$X_{k|k}=X_{k|k-1}+K_{k}(Y_{k}-y_{k|k-1})$$
(18)$$P_{k|k}=(I-K_{k}H)P_{k|k-1}$$

## 3. Improved Strong Tracking Cubature Kalman Filtering Algorithm

In order to fully mitigate the positioning error of the UWB positioning system, it is necessary to identify the main errors in the positioning error of the system. It is then necessary to identify whether the observations are NLOS observations or not at the current moment. After the identification result is obtained, different strategies are adopted for different cases. In the LOS environment, the main errors of the UWB positioning system are the prediction dynamics model errors, so the dynamics model errors should be mitigated. In the NLOS environment, the main positioning errors of the UWB positioning system are NLOS errors, so it is necessary to mainly mitigate NLOS errors.

First, the proposed algorithm reconstructs the observations to obtain the new observations, so as to mitigate the observation noise and NLOS errors in the original observations. Second, in the process of filtering the reconstructed observations to obtain the positioning results, different filtering methods are used according to the NLOS identification results. In the LOS environment, fading filtering is used to mitigate the predicted kinetic model errors. And in the NLOS environment, inverse fading filtering is used to emphasize the historical observation information and thus mitigate the NLOS errors. Finally, the main errors in the UWB positioning system are fully mitigated at each moment.

### 3.1. Reconstruction and Identification of NLOS Observations

According to the characteristics of LOS observation noise and NLOS observation noise, the variance of observation noise in NLOS environment is much larger than the variance of LOS observation noise and its mean value is also much larger than the mean value of LOS observation noise, so as to carry out the identification of NLOS observation. For the estimation of the associated judgment statistic, the new observations are obtained by reconstructing the original observations, and then the corresponding judgment statistic is operated between the original observation and the reconstructed observation. 

The reconstruction observations are estimated as follows. First, the positioning results estimated by the predictive dynamics model of CKF are combined with the positioning results estimated by least squares to obtain the weighted positioning results. This result is then used to compute accordingly with the anchor node coordinates to obtain the primary reconstructed observations. Finally, based on the identification result of primary NLOS observations, the positive characteristics of NLOS observation errors and the relative smoothness of tag motion state, the primary reconstructed observations are corrected by using the original observation and the primary reconstructed observation to obtain the secondary reconstructed observations.

Since the original observation is reconstructed twice, the NLOS identification is performed twice accordingly. The reconstruction and identification algorithm for the observations is as follows.

#### 3.1.1. Primary Reconstructed Observation

At moment *k*, the observations are combined into different subgroups, and the number of anchor nodes in each subgroup is three. Assuming that there are $n_{y}$ anchor nodes, the division results in $C_{n_{y}}^{3}=\beta$ subgroups. In this paper, the three anchor nodes are used as a subgroup to estimate the tag locations $MS_{i},i=1,2,\ldots ,\beta +1$ by the least squares algorithm. Where $MS_{\beta +1}$ is the predicted positioning result. The new tag position ${\hat{M}}_{k}$ are estimated by a weighted combination of $MS_{i},i=1,2,\ldots ,\beta +1$, where the weights are obtained from the data residuals between the actual observations and the corresponding observations obtained by $MS_{i}$ through h(·).
(19)$$Z_{MS_{i}}=h(MS_{i}),i=1,2,\ldots ,\beta +1$$
where $Z_{MS_{i}}\in R^{1\times n_{y}}$ and $MS_{i}\in R^{2\times 1}$.
(20)$$W_{Z_{MS_{i}}}=\frac{1}{\sum_{j=1}^{n_{y}}\left|Y_{k,j}-Z_{MS_{i},j}\right|},i=1,2,\ldots ,\beta +1$$
where $Y_{k}\in R^{1\times n_{y}}$ are the observation at moment *k*, $Y_{k,j}$ and $Z_{MS_{i},j}$ are denoted as the elements of column *j* of $Y_{k}$ and $Z_{MS_{i}}$, respectively.
(21)$$W_{i}=W_{Z_{MS_{i}}}/\sum_{j=1}^{\beta +1}W_{Z_{MS_{j}}},i=1,2,\ldots ,\beta +1$$
(22)$${\hat{M}}_{k}=\sum_{i=1}^{\beta +1}W_{i}MS_{i}$$

The estimation of the true observations is performed from the weighted positioning result ${\hat{M}}_{k}$ obtained at moment *k*. The reconstructed observations are obtained:(23)$${\hat{y}}_{1,k}=h({\hat{M}}_{k})$$
where the restriction on ${\hat{y}}_{1,k}$ is:(24)$$\{\begin{array}{c} {\hat{y}}_{1,ki}=Y_{ki},{\hat{y}}_{1,ki}\geq Y_{ki}\\ {\hat{y}}_{1,ki}={\hat{y}}_{1,ki},{\hat{y}}_{1,ki}<Y_{ki} \end{array}$$

This operation is to mitigate the effect of NLOS errors in reconstructing the observations.

#### 3.1.2. Primary NLOS Identification

The observation noise variance is estimated from the primary reconstructed observation ${\hat{y}}_{1,k}$ obtained above, then:(25)$${\hat{R}}_{1,ki}={\left(Y_{ki}-{\hat{y}}_{1,ki}\right)}^{2}$$
where *i* denotes the *i*th column of the corresponding vector. 

From the data obtained above, the NLOS identification of the observations at moment *k* is given by:(26)$$\mathrm{Condition}\;1:\;{\hat{R}}_{1,ki}\{\begin{array}{c} \geq {\hat{R}}_{1,ki,thr},\;\mathrm{NLOS}\;\mathrm{Propagation}\;\mathrm{state}\\ <{\hat{R}}_{1,ki,thr},\;\mathrm{LOS}\;\mathrm{Propagation}\;\mathrm{state} \end{array}$$
where the relevant threshold is calculated as follows:(27)$$r_{1,ki}=|Y_{ki}-{\hat{y}}_{1,ki}|$$
(28)$${\bar{r}}_{1,ki,thr}=\alpha {\bar{r}}_{1,(k-1)i,thr}+(1-\alpha )r_{1,ki}$$
where ${\bar{r}}_{1,ki,thr}$ is the data for which $r_{1,ki}$ is smoothed and denoised. α is the weighting factor of the historical observations from the exponentially weighted moving average (EWMA), and α=0.99 is taken to remove the effect of NLOS errors as much as possible. Then: (29)$${\hat{R}}_{1,kj,thr}={\bar{r}}_{1,kj,thr}^{2}+3{\hat{\sigma }}_{los}^{2}$$
where ${\hat{\sigma }}_{los}^{2}$ is the fitted observation noise variance in the LOS environment, whose estimation is covered below. 

When an NLOS observation exists at moment *k*, the reconstruction and identification of the observation is affected by the NLOS error. In order to mitigate the effect, the NLOS identification results determined by condition 1 are corrected according to the positive property of NLOS errors. The specific algorithm is as follows.

Assume that the identification result of condition 1 for $Y_{ki}$ is the NLOS observation, the identification result for $Y_{(k-1)i}$ is the NLOS observation, and the identification result for $Y_{(k-2)i}$ is the LOS observation. Then the algorithm flow is as follows.
(30)$$\{\begin{array}{c} Flag_{1,k,i}=1,\;\;\;\;△y1_{ki}\geq \Delta y1_{ki,thr}\\ Flag_{1,k,i}=Flag_{1,k-1,i},\;△y1_{ki}<\Delta y1_{ki,thr} \end{array}$$
where $Flag_{1,k,i}$ takes the value of 0 when condition 1 determines that $Y_{ki}$ is an LOS observation. then: (31)$$Flag_{1,k,i}=\{\begin{array}{c} 1,\;\mathrm{NLOS}\;\mathrm{Propagation}\;\mathrm{state}\\ 0,\;\;\mathrm{LOS}\;\mathrm{Propagation}\;\mathrm{state} \end{array}$$

The relevant data in (30) are obtained as follows. First, the fitted estimate of the true change in the observations is given by:

The amount of change of $Y_{ki}$ and $Y_{(k-1)i}$ is:(32)$$△y1_{ki}=Y_{ki}-Y_{(k-1)i}$$

EWMA smoothing is performed on $△y1_{ki}$ to obtain the fitted variation $△y1_{ki}^{\prime }$ of $△y1_{ki}$ under the LOS environment at moment *k*, which is:(33)$$△y1_{ki}^{\prime }=\alpha △y1_{(k-1)i}^{\prime }+(1-\alpha )△y1_{ki}$$
where α is taken as 0.9 in order to fit the tendency of the observations under the LOS environment as much as possible.

Second, the fitted estimation of the observation noise in the LOS environment is performed according to the absolute value of difference between $△y1_{ki}$ and $△y1_{ki}^{\prime }$.
(34)$$\Delta y2_{ki}=|△y1_{ki}-△y1_{ki}^{\prime }|$$

The EWMA algorithm is used to mitigate the effect of NLOS errors in $\Delta y2_{ki}$.
(35)$${\hat{\sigma }}_{los,ki}=\alpha {\hat{\sigma }}_{los,(k-1)i}+(1-\alpha )\Delta y2_{ki}$$
where α takes the value of 0.99.

Therefore, according to the positive characteristic of NLOS errors, the 1σ criterion is adopted for the threshold calculation:(36)$$\Delta y1_{ki,thr}=△y1_{ki}^{\prime }+{\hat{\sigma }}_{los,ki}$$

#### 3.1.3. Secondary Reconstructed Observation 

The correction for the reconstructed observations according to the identification result obtained from the Equation (31) is given by:

If the identification for $y_{ki}$ is an NLOS observation, the correction for ${\hat{y}}_{1,ki}$ is:(37)$${\hat{y}}_{2,ki}=\{\begin{array}{c} {\hat{y}}_{1,ki},\;\;\;\;{\hat{y}}_{1,ki}-Y_{ki}\geq △d1_{ki}^{\prime }+{\hat{\sigma }}_{los,ki}\\ w_{1}{\hat{y}}_{1,ki}+w_{2}Y_{ki},\;{\hat{y}}_{1,ki}-Y_{ki}<△d1_{ki}^{\prime }+{\hat{\sigma }}_{los,ki} \end{array}$$

The Equation (37) is a reliability judgment on ${\hat{y}}_{1,ki}$, and operates on ${\hat{y}}_{1,ki}$ accordingly. Where: (38)$$w_{1}=\Delta 1_{k,i}/(\Delta 1_{k,i}+\Delta 2_{k,i})$$
(39)$$w_{2}=\Delta 2_{k,i}/(\Delta 1_{k,i}+\Delta 2_{k,i})$$
(40)$$\Delta 1_{k,i}=|Y_{k,i}-Y_{k-1,i}|$$
(41)$$\Delta 2_{k,i}=|{\hat{y}}_{k,i}-Y_{k-1,i}|$$

If the NLOS identification of $Y_{ki}$ results in an LOS observation, there are two cases:

If the observation $Y_{kj},\;j\neq i$ at moment *k* is an NLOS observation, then the correction for ${\hat{y}}_{1,ki}$ is given by:(42)$${\hat{y}}_{2,ki}=Y_{ki}$$

If the observation $Y_{kj},\;j\neq i,j=1,2,\ldots ,n_{y}$ at moment *k* is an LOS observation, the correction for ${\hat{y}}_{1,ki}$ is:(43)$${\hat{y}}_{2,ki}=w_{i}Y_{ki}+(1-w_{i}){\hat{y}}_{1,ki}$$
(44)$$w_{i}=R_{1,ki}/((\sum_{j=1}^{n_{y}}R_{1,kj})/n_{y}),i=1,2,\ldots ,n_{y}$$
where:(45)$$\{\begin{array}{c} w_{i}=0.9,w_{i}\geq 0.9\\ w_{i}=w_{i},w_{i}<0.9 \end{array}$$

#### 3.1.4. Secondary NLOS Identification

Because the primary NLOS identification is easily affected by the NLOS errors, the secondary reconstructed observations are used to correct the identification result. Since the mean value of NLOS errors is much larger than the mean value of LOS noise, the difference between the secondary reconstructed observations and the original observations is used to correct the primary NLOS identification results. The algorithm flow is as follows.

According to the primary NLOS identification result in the Equation (31), this identification result is corrected as follows:(46)$$\{\begin{array}{c} r_{2,ki}>r_{2,ki,thr},\;\mathrm{NLOS}\;\mathrm{Propagation}\;\mathrm{state}\\ r_{2,ki}\leq r_{2,ki,thr},\;\mathrm{LOS}\;\mathrm{Propagation}\;\mathrm{state} \end{array}$$

The calculation procedure for the parameters in the Equation (46) is as follows:(47)$$r_{2,ki}={\hat{y}}_{2,ki}-Y_{ki}$$
(48)$$r_{2,ki,thr}=\alpha r_{2,(k-1)i,thr}+(1-\alpha )r_{2,ki}+{\hat{\sigma }}_{los,ki}$$
where α is taken as 0.99. 

The flowchart of NLOS identification in the algorithm of this paper is shown in Figure 2.

### 3.2. Estimation of Observation Noise Variance

The square of the difference between the original observations and the primary reconstructed observations is used as the estimation result of the observation noise variance. Therefore, the estimation of its observation noise variance for $y_{ki}$ is performed as follows:(49)$${\hat{r}}_{ki}={\left(Y_{ki}-{\hat{y}}_{1,ki}\right)}^{2},i=1,2,\ldots ,n_{y}$$

Perform sliding window smoothing on ${\hat{r}}_{ki}$ as follows:(50)$${\hat{R}}_{ki}=\frac{1}{\lambda }\sum_{j=k-\lambda }^{k}{\hat{r}}_{ji}$$
where the window length λ is 2. 

Therefore, the estimate of the observation noise covariance in the CKF are obtained as follows:(51)$${\hat{R}}_{k}=\left[\begin{array}{cccc} {\hat{R}}_{k1} & 0 & \cdots & 0\\ 0 & {\hat{R}}_{k2} & \cdots & 0\\ \vdots & \vdots & \ddots & \vdots \\ 0 & 0 & \cdots & R_{kn_{y}} \end{array}\right]$$
where ${\hat{R}}_{k}\in R^{n_{y}\times n_{y}}$.

Considering the susceptibility of ${\hat{R}}_{k}$ to NLOS errors, the scheme of adaptive estimation of the observation noise covariance in the ACKF is applied to the correction of ${\hat{R}}_{k}$. Therefore: (52)$${\hat{R}}_{k}=\alpha {\hat{R}}_{k}+(1-\alpha )(e_{k}e_{k}{}^{T}-HP_{k|k-1}H^{T})$$
where α∈[0.9,1], and α is taken to be 9.3 in this paper.

### 3.3. Estimation of Predictive Dynamics Model Error Covariance

Weighting Adaptive sliding window is used to estimate the predictive dynamics model error covariance, where the adaptive estimation is performed according to the cubature points in the CKF. The algorithm is as follows.

#### 3.3.1. Adaptive Estimation

The predicted state vector $X_{k|k-1}$ and the filtered state vector $X_{k|k}$ of the system at moment *k* are known. 

There is a deviation in the predicted dynamics model in the actual scenario. Assume that the deviation is $W_{k}^{{}^{\prime }}$ and the observation noise is zero at moment *k*.
(53)$$X_{k}=X_{k|k-1}+W_{k}^{{}^{\prime }}$$
where $X_{k}$ is the true state vector of the tag. Assume
(54)$$E(W_{k}^{{}^{\prime }})=S_{k}$$

Therefore, the covariance of $W_{k}^{{}^{\prime }}$ is:(55)$$\Sigma_{W_{k}^{\prime }}=E\left[(X_{k}-X_{k|k-1}-S_{k}){\left(X_{k}-X_{k|k-1}-S_{k}\right)}^{{}_{T}}\right]$$

According to the Equations (4)–(6), the approximate estimation of the Equations (54) and (55), gives:(56)$$S_{k}=E(X_{k}-X_{k|k-1})=\sum_{i=1}^{m}\left(X_{k}-Xf_{i,k|k-1}\right)$$
(57)$$\Sigma_{W_{k}^{\prime }}=\frac{1}{m}\sum_{i=1}^{m}(X_{k}-Xf_{i,k|k-1}-S_{k}){\left(X_{k}-Xf_{i,k|k-1}-S_{k}\right)}^{{}_{T}}$$
where $Xf_{i,k|k-1}$ is the predicted state vector cubature point at moment *k*. Thus, an approximate estimate ${\hat{Q}}_{1,k}\approx {\hat{\Sigma }}_{W_{k}^{\prime }}$ of the predicted kinetic model deviation covariance can be obtained. $X_{k|k}$ is used in this paper to approximate $X_{k}$ instead, so:(58)$${\hat{S}}_{k}=\sum_{i=1}^{m}\left(X_{k|k}-Xf_{i,k|k-1}\right)=X_{k|k}-X_{k|k-1}$$
(59)$${\hat{Q}}_{1,k}=\frac{1}{m}\sum_{i=1}^{m}(X_{k|k}-Xf_{i,k|k-1}-{\hat{S}}_{k}){\left(X_{k|k}-Xf_{i,k|k-1}-{\hat{S}}_{k}\right)}^{{}_{T}}$$

#### 3.3.2. Estimation of Weighted Sliding Window

Because $X_{k|k}$ is used in the adaptive estimation of ${\hat{Q}}_{1,k}$, the NLOS errors and observation noise have a significant effect on the estimation ${\hat{Q}}_{1,k}$ of the predictive dynamics model errors covariance. Since the tag motion state can be considered to be approximately smooth for a short period of time, the weighted sliding window is then used to correct the predictive dynamics model error covariance ${\hat{Q}}_{1,k}$. The specific algorithm is as follows:(60)$$w_{ri}=1/|{\hat{S}}_{r,i}|$$
(61)$$w_{ri}=w_{ri}/\sum_{j=k-\beta }^{k}w_{ji},i=1,2,\ldots ,n_{x}\;r=k-\beta +1,\ldots ,k$$
where ${\hat{S}}_{r,i}$ is the element of the *i*th row of ${\hat{S}}_{r}$, and β is the length of the window, so $w_{k}\in R^{n_{x}\times \beta }$.

Therefore, the correction to ${\hat{Q}}_{1,k}$ is:(62)$${\hat{Q}}_{1,k}=\sum_{i=k-\beta +1}^{k}{\left(w_{k}\right)}_{i+\beta -k}.\times {\hat{Q}}_{1,i}$$
(63)$${\hat{Q}}_{1,k}=\alpha {\hat{Q}}_{1,k-1}+(1-\alpha ){\hat{Q}}_{1,k}$$
where ${\left(w_{k}\right)}_{i+\beta -k}$ is the i+β−k th column vector of $w_{k}$. α is the weighting factor of EWMA, which is taken as 0.99.

From the computational procedure of ${\hat{Q}}_{1,k}$, and according to the smoothness of the tag movement, we trade-off the computational complexity with higher accuracy and use the estimated result ${\hat{Q}}_{1,k}$ at moment *k* for moment *k* + *1*. Then:(64)$$Q_{k+1}={\hat{Q}}_{1,k}$$

### 3.4. Improved Strong Tracking Cubature Kalman Filter

The enhancement of the strong tracking cubature Kalman filter that renders it suitable for UWB positioning system lies in two specific improvements.

#### 3.4.1. Substitution of Cholesky Decomposition

When the filter error covariance matrix and the prediction error covariance matrix are estimated, the arithmetic process may lead to the fact that the filter error covariance matrix is often not a positive definite matrix. The covariance does not then satisfy the requirements of Cholesky decomposition, which in turn leads to instability of the filtering process. Therefore, this paper uses singular value decomposition instead of Cholesky decomposition to improve the stability of the proposed algorithm.

For the Equations (4) and (9), the singular value decomposition is performed instead as follows:(65)$$P_{k-1|k-1}=U_{k-1|k-1}\Lambda_{k-1|k-1}U_{k-1|k-1}$$
where $\Lambda_{k-1|k-1}=D_{k-1|k-1}{}^{2}$, then:(66)$$P_{k-1|k-1}=U_{k-1|k-1}D_{k-1|k-1}D_{k-1|k-1}U_{{}_{k-1|k-1}}^{T}$$

Then: (67)$$S_{k-1|k-1}=U_{k-1|k-1}D_{k-1|k-1}$$

Similarly, there are:(68)$$P_{k|k-1}=U_{k|k-1}\Lambda_{k|k-1}U_{k|k-1}$$
(69)$$P_{k|k-1}=U_{k|k-1}D_{k|k-1}D_{k|k-1}U_{k|k-1}^{T}$$
(70)$$S_{k|k-1}=U_{k|k-1}D_{k|k-1}$$

#### 3.4.2. Solving for Fading Factor

From the secondary NLOS identification results mentioned above, a multi-factor combination of fading and inverse fading is used to correct the prediction vector covariance. This makes the utilization for historical observations more consistent with the judgment of the main errors. The specific algorithm is as follows.

The modification of the Equation (8) is given by:(71)$${\hat{P}}_{k|k-1}={\hat{\varepsilon }}_{k}(\frac{1}{m}\sum_{i=1}^{m}Xf_{i,k|k-1}{\left(Xf_{i,k|k-1}\right)}^{T}-X_{k|k-1}{\left(X_{k|k-1}\right)}^{T})+Q_{k}$$
where ${\hat{\varepsilon }}_{k}$ is the fading factor for the predicted state vector covariance, which is calculated as follows.
(72)$$M_{k}=H_{k}F_{k}P_{k-1|k-1}F_{k}{}^{T}H_{k}{}^{T}$$
(73)$$N_{k}=P_{V_{Z_{k|k-1}}}-H_{k}Q_{k}H_{k}{}^{T}-R_{k}$$
where $P_{V_{Z_{k|k-1}}}$ is the covariance matrix of the data residuals $e_{k}$, estimated by a sliding window method similar to the Sage filter. $H_{k}$ is the result of linearization on $h(X_{k|k-1})$. Then:(74)$$H_{k}=\frac{dh\left(x\right)}{dx}{|}_{x=X_{k|k-1}}$$
(75)$$P_{V_{Z_{k|k-1}}}=\frac{1}{\eta }\sum_{i=k-\eta +1}^{k}e_{i}e_{i}^{T}$$
where η is the window length and can be taken as 5.

The traditional single fading factor $\varepsilon_{k}=tr(N_{k})/tr(M_{k})$ cannot fully utilize the performance because it has the same adjustment weight for each state channel of $P_{k|k-1}$. Multiple fading factors assign different fading weights to each state channel, which can better adjust the gain matrix adaptively and enhance the stability of the filtering when the system model is mismatched. The proposed algorithm uses different calculation models for fading factors based on the NLOS identification results and the fading factor. The fading factor is calculated as follows:(76)$$\varepsilon_{k,i}=\left|N_{k}(i,i)/M_{k}(i,i)\right|,i=1,2,\ldots ,n_{x}$$
where $\varepsilon_{k,i}$ denotes the *i*th row element of the fading factor vector $\varepsilon_{k}$ at moment *k*, and $N_{k}(i,i)$ and $M_{k}(i,i)$ denote the *i*th diagonal element of $N_{k}$ and $M_{k}$.

The correction for the above fading factor is given by:

If the identification of the observation $Y_{ki},i=1,2,\ldots ,n_{y}$ results in an LOS observation, then:(77)$${\hat{\varepsilon }}_{k,i}=\{\begin{array}{c} 1,\varepsilon_{k,i}\leq \mu \\ \varphi ,\varepsilon_{k,i}>\mu \end{array}$$
where φ ∈ (1, 10), which indicates the emphasis on $Y_{ki}$, and φ is taken as 2.7 in this paper.

If the identification of observation $Y_{ki},i=1,2,\ldots ,n_{y}$ results in an NLOS observation, then:(78)$${\hat{\varepsilon }}_{k,i}=\{\begin{array}{c} 1,\varepsilon_{k,i}\leq \mu \\ \mu /\varepsilon_{k,i},\varepsilon_{k,i}>\mu \end{array}$$

If the identification of observation $Y_{ki},i=1,2,\ldots ,n_{y}$ results in an LOS observation and there are the NLOS observations in $Y_{k}$, then:(79)$${\hat{\varepsilon }}_{k,i}=1$$
where μ is the determination limit between the observation noise and the predicted kinetic model errors or NLOS errors, which is taken as 3 in this paper.

The flow chart of the proposed ISTCKF is shown in Figure 3, where the details are given in Table 1. The main idea of Figure 3 is as follows: The proposed algorithm in this paper obtains another observation by reconstructing the observations. It then uses the two observations and the difference of the statistical properties between the LOS noise and the NLOS noise for NLOS identification. Then, the main error sources of the UWB positioning system at the current moment are judged by the NLOS identification results and the fading factor calculated, and then the corresponding strategies are used for the mitigation of the corresponding errors.

The steps of the proposed ISTCKF are shown in Table 1.

## 4. Experimental Analysis

### 4.1. Simulation and Analysis of the Proposed Algorithm

In this section, MATLAB 2019b is used to simulate and analyze the positioning algorithm in this paper. The simulation scenario and parameters are set as follows: four anchor nodes are deployed within 20 m × 20 m. The coordinates of the anchor nodes are BS1 (0 m, 0 m), BS2 (0 m, 20 m), BS3 (20 m, 0 m) and BS4 (20 m, 20 m). The tags were moved sequentially in a uniform linear motion between five points from (0 m, 0.1 m), (6 m, 0.1 m), (6 m, 4.1 m), (0 m, 4.1 m), (0 m, 0.1 m). LOS Gaussian noise and simulated continuous pedestrian NLOS errors were added to the observations, which are shown in Figure 2.

The LOS noise N(0,2cm) added to the observations. From Figure 1: The value of the pedestrian NLOS error is around 90 to 110 cm. Therefore, the simulation of the NLOS error is performed with a uniform distribution U(90 cm, 120 cm). The simulation of the observation was performed for the cases of 1, 2, 3 and 4 NLOS noises at the same moment with full consideration of the appearance of the NLOS errors. The observation results are shown in Figure 4.

The EKF, CKF and UKF are susceptible to predictive dynamics model errors. Therefore, they have a delay in tracking the tag motion state when the tag motion state changes. Therefore, the proposed algorithm is compared with ACKF [38], STCKF [39], and RSTCKF for positioning tracking effect to verify the effectiveness of the proposed algorithm. The scheme of the adaptive estimation of observation noise in ACKF is applied to STCKF, and RSTCKF is obtained. The experimental simulation scenario is shown in Section 4.1 above. The observation noise covariance in STCKF is set as a diagonal matrix, and the values of the diagonal elements in the diagonal matrix are taken randomly from 0 to 10. The predictive dynamics model error covariance in STCKF and RSTCKF is taken according to the dynamics, where the random acceleration is taken as 1. These algorithms were subjected to 100 Monte Carlo simulations to ensure the stability and effectiveness of the proposed algorithms. The simulation results are as follows.

From Figure 5, it can be seen that the NLOS errors of the reconstructed observations are considerably improved. When there are more LOS observations, it can be seen that the NLOS errors mitigation effect of the reconstructed observations is obvious. In the case of fewer LOS observations, the NLOS errors mitigation of the reconstructed observations is poor. And it can be seen from Figure 3 that the reconstructed observation in the LOS environment also has a mitigation effect on the observation noise.

The single simulation positioning result of the Figure 6 is the Figure 7. Therefore, Figure 7 better represents the actual localization of each algorithm compared to Figure 4. Figure 6 can prove the stability and effectiveness of each algorithm. From Figure 6, it can be observed that the proposed algorithm performs much better than the other algorithms in mitigating the NLOS error, but worse than the other algorithms in mitigating the error of the predictive dynamics model. This is because the proposed algorithm uses the corresponding estimation method in ACKF to correct the original estimation result observation noise covariance. Therefore, it incorporates the prediction dynamics model error into the estimation of observation noise when the motion state of the tag changes. Ultimately, the observation noise covariance at this time is large, which places relatively more emphasis on the predictive dynamics model.

Table 2 shows the data results of the information in Figure 6 and Figure 8. The overall average RMSE error results in Table 2 show that the proposed algorithm has better positioning results. Compared with ACKF, STCKF and RSTCKF, the positioning accuracy of the proposed algorithm is improved by 55%, 79% and 71%, respectively. From the NLOS positioning errors in Table 2, it can be seen that the proposed algorithm mitigates the NLOS errors very well. In the above cases, the mitigating effect of the proposed algorithm for the NLOS errors in UWB positioning is much better than the other algorithms. Compared with ACKF, STCKF and RSTCKF, the proposed algorithm is more effective in mitigating the NLOS errors in UWB positioning and also ensures mitigation of the prediction dynamics model error.

### 4.2. Analysis of Field Experiment Results

To validate the effectiveness of the proposed algorithm, the DWM1000 UWB positioning module was selected for field experiments. The experimental site is located in the hall on the first floor of the experiment building, which has four columns in the center. In this positioning scenario, the tags move around the columns so that the columns cause NLOS interference with the observations. In this experiment, four anchor nodes with one tag were set up, and the handheld tag moved at a relatively smooth speed according to a preset trajectory. During this period, the system performed positioning every 0.2 s and stopped data acquisition when it reached the end point, where the height of the anchors was 215 cm. In the real scenario, the ground is relatively smooth, and the tag started from (1076 cm, 102 cm) and moved around a rectangle with a length of 1106 cm and a width of 963 cm. The specific experimental scenario is shown in Figure 9. The actual observations are shown in Figure 10. The effect of the reconstructed observation on the mitigation effect of the observation noise is shown in Figure 10, and the effects of ACKF, STCKF, RSTCKF and the proposed positioning algorithm in this paper are shown in Figure 11.

During the experiment, the signal sometimes became obscured by the columns while being transmitted between the receiver and the tag. Therefore, the UWB observation is affected by the NLOS errors. At the same time, there is an error between the moving trajectory of the tag and the expected trajectory, which makes the covariance matrix of the predicted information unknown. Through the above field experiments, the relevant results are obtained, i.e., Figure 10, Figure 11, Figure 12 and Figure 13.

From Figure 11 and Figure 12, it can be seen that the positioning error of the tag increases when the observation information is affected by the NLOS errors. The proposed algorithm handles the UWB positioning error well, especially for the NLOS errors, so the optimal positioning results of the proposed algorithm can be seen from the figures. The average positioning errors of STCKF, ACKF, RSTCKF and ISTCKF are 22.04 cm, 14.57 cm, 13.88 cm and 9.86 cm, respectively. Compared with STCKF, ACKF and RSTCKF, the positioning accuracy of the proposed algorithm is improved by 55.2%, 32.3% and 28.9%, respectively.

From Figure 13, it can be seen that the velocity component in the filtered positioning results of this proposed algorithm remains relatively stable compared to other algorithms. The result is in line with the field experiment, in which the tag moves in a relatively smooth state. Therefore, the proposed algorithm demonstrates enhanced stability.

## 5. Discussion

In the process of tracking tags, the Kalman filtering algorithm is mainly affected by the NLOS errors and the predictive dynamics model errors. The proposed algorithm performs the NLOS identification, and judges the main errors in the filtering algorithm in the UWB positioning system at the current moment, and thus adopts the corresponding strategy to mitigate the main error. Compared with STCKF, ACKF and RSTCKF, the positioning accuracy of the proposed algorithm is improved by 55.2%, 32.3% and 28.9%, respectively. The field experimental results show that the tag trajectory obtained by the proposed algorithm is closer to the real trajectory than the tag trajectory obtained by other algorithms, and its positioning error is significantly smaller.

The main positioning errors of the UWB filtering system, namely the predictive dynamics model errors and NLOS errors, are mitigated by the improved strong tracking filtering. The main idea is to mitigate the predictive dynamics model errors by focusing on the current observations, and to mitigate the NLOS errors by focusing on the historical observations. Therefore, it is necessary to determine the trust level of the observations at the current moment, and thus the NLOS identification of the observations is required. The proposed algorithm identifies the LOS/NLOS state of the signal propagation path by the reconstructed observations. And this identification scheme performs dynamic identification of UWB signals. However, the proposed algorithm has some drawbacks, such as the effect of the NLOS errors in the estimation of the reconstructed observations, and the judgment of the trustworthiness of the observations in the solution of the correlation fading factors. The proposed algorithm only makes a simple judgment on the bounds between the observation noise and NLOS errors or the prediction dynamics model errors. Therefore, it will produce some misjudgment on the trust degree of the observations at the current moment, which will affect the positioning accuracy. Therefore, the judgment threshold concerning the trustworthiness of the observations can be considered comprehensively for adaptive localization to enable a more reasonable determination of the relevant thresholds.

## 6. Summary

In order to meet the high-accuracy requirements of the UWB positioning systems in mobile scenes, CKF is applied to positioning techniques. However, problems such as predictive dynamics model errors and NLOS errors significantly affect the positioning accuracy of the system. In order to fully mitigate the positioning error, firstly, the proposed algorithm processes the observations in real time, and thus obtains the reconstructed observations. Secondly, the NLOS identification is performed according to the difference in statistical properties between the LOS observations noise and the NLOS noise. Finally, according to the identification results and the reconstructed observations, the main positioning error at the current moment is fully mitigated by the ISTCKF. The simulation results and the field experimental results show that:The proposed algorithm obtains the corresponding ISTCKF algorithm at the current moment by reconstructing the observations and the NLOS identification results. It then uses this ISTCKF to obtain the positioning result. Compared with STCKF, ACKF and RSTCKF, the positioning accuracy of the proposed algorithm is improved by 55.2%, 32.3% and 28.9%, respectively. Therefore, the proposed algorithm significantly improves the positioning accuracy.The simulation and experiments show that the proposed algorithm is effective in mitigating the NLOS errors. And since the proposed algorithm includes NLOS identification, it is better suited to deal with UWB positioning errors. Therefore, the proposed algorithm offers enhanced positioning accuracy and relative stability.

## Figures and Tables

**Figure 1 sensors-23-07463-f001:**
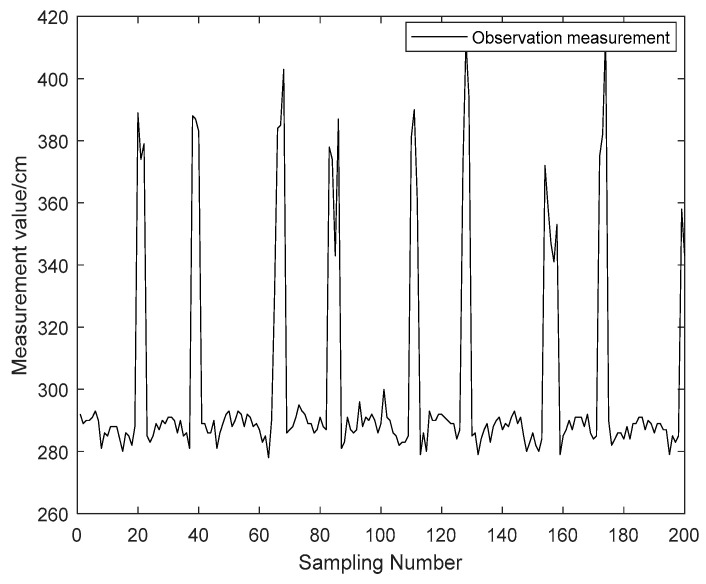
NLOS errors caused by a pedestrian.

**Figure 2 sensors-23-07463-f002:**
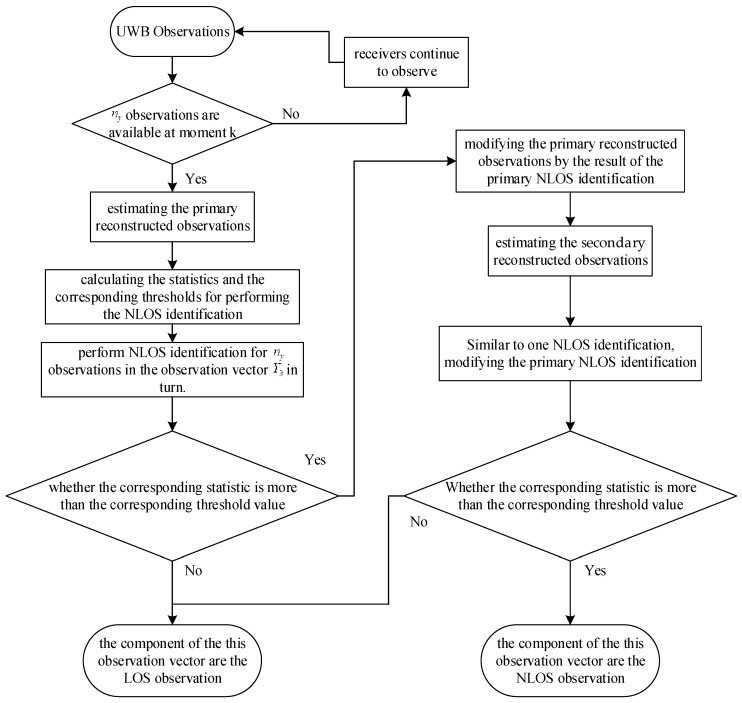
The NLOS recognition scheme based on reconstructed observations.

**Figure 3 sensors-23-07463-f003:**
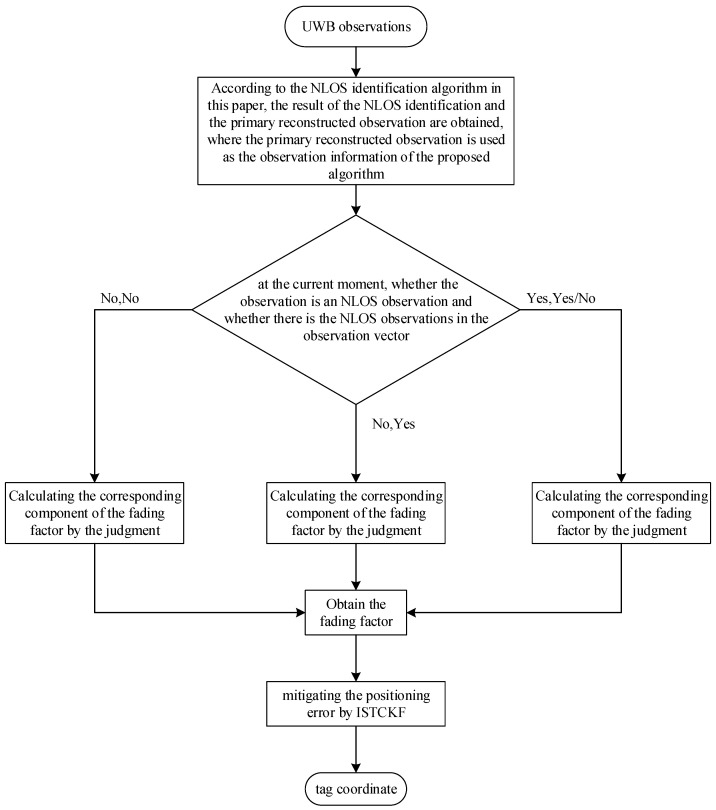
Flow chart of the proposed ISTCKF.

**Figure 4 sensors-23-07463-f004:**
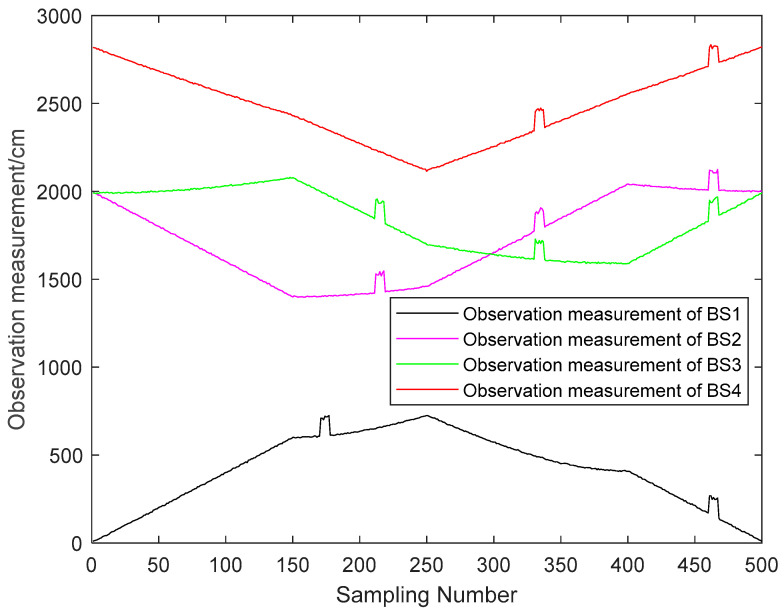
The observation measurement.

**Figure 5 sensors-23-07463-f005:**
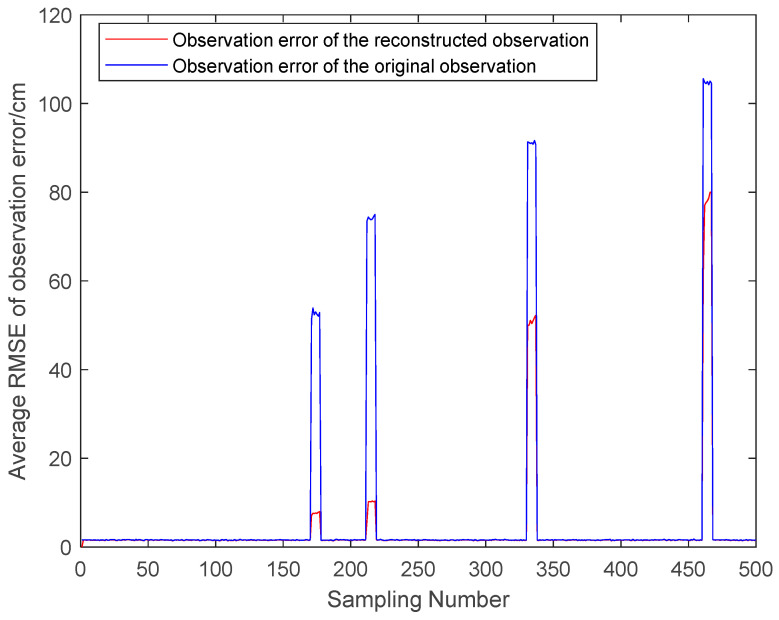
The observation error.

**Figure 6 sensors-23-07463-f006:**
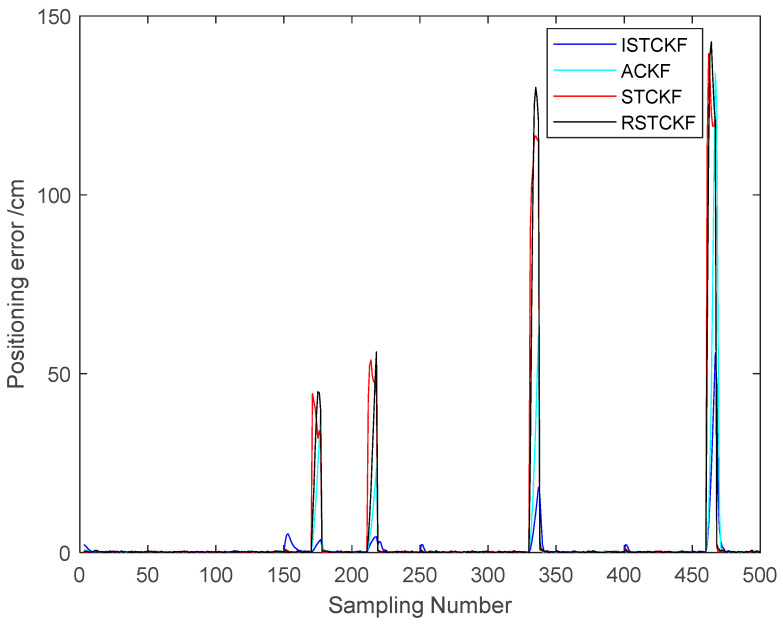
The positioning error of the algorithms.

**Figure 7 sensors-23-07463-f007:**
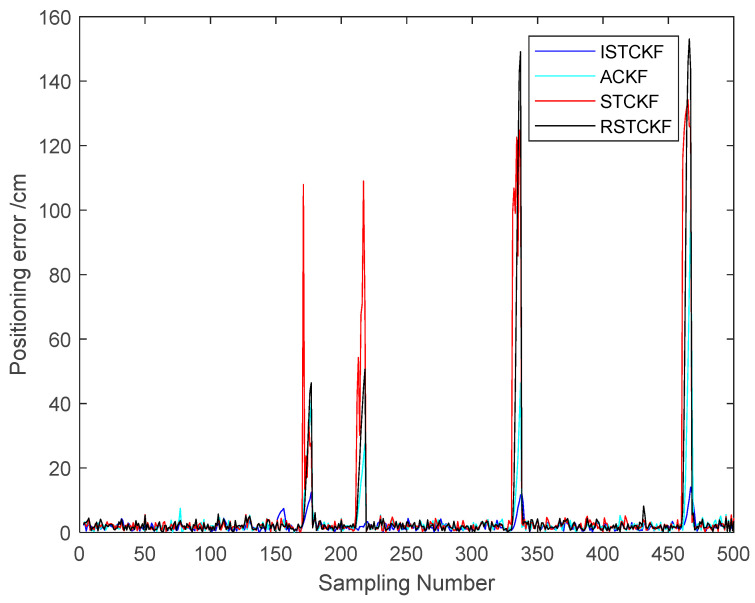
The positioning error of the algorithms in a single pass.

**Figure 8 sensors-23-07463-f008:**
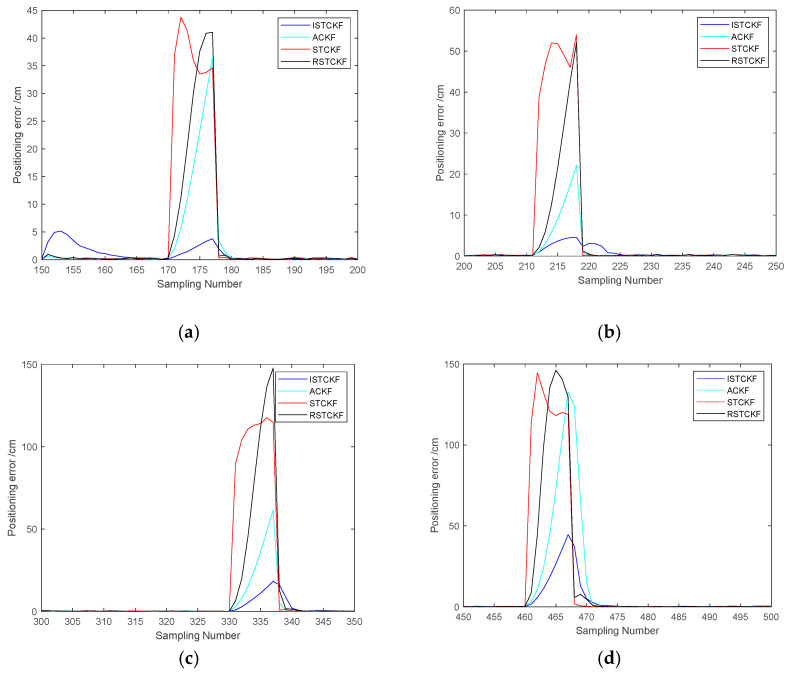
The positioning error of the algorithms. (**a**) The positioning error when 1 NLOS observations exist at the same time; (**b**) The positioning error when 2 NLOS observations exist at the same time; (**c**) The positioning error when 3 NLOS observations exist at the same time; (**d**) The positioning error when 4 NLOS observations exist at the same time.

**Figure 9 sensors-23-07463-f009:**
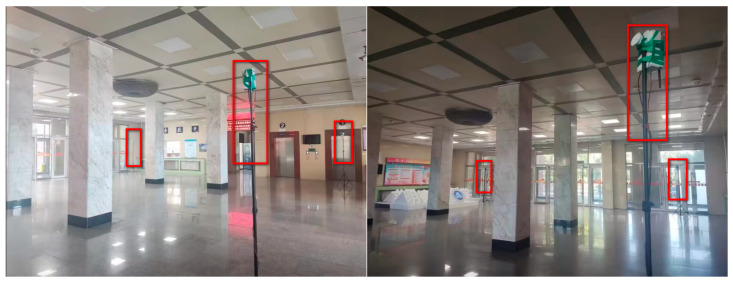
The experimental scenario. Note: The red frames in the Figure 9 are marked for the anchors in the experimental site.

**Figure 10 sensors-23-07463-f010:**
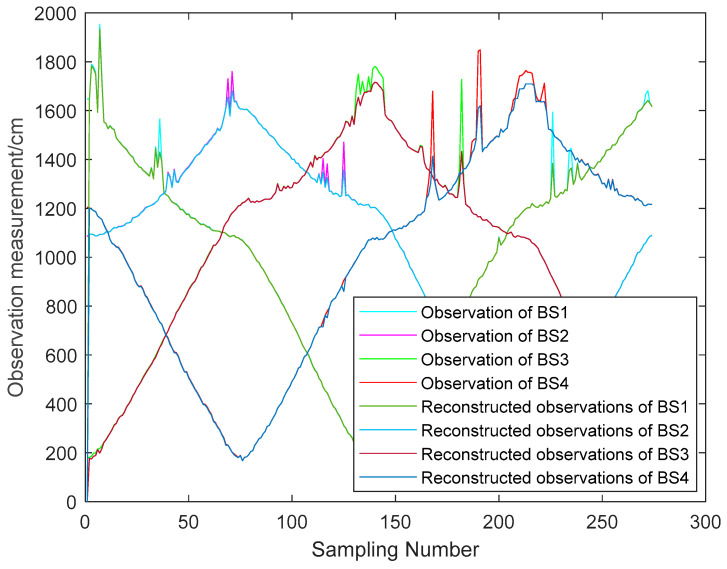
The original and reconstructed observations.

**Figure 11 sensors-23-07463-f011:**
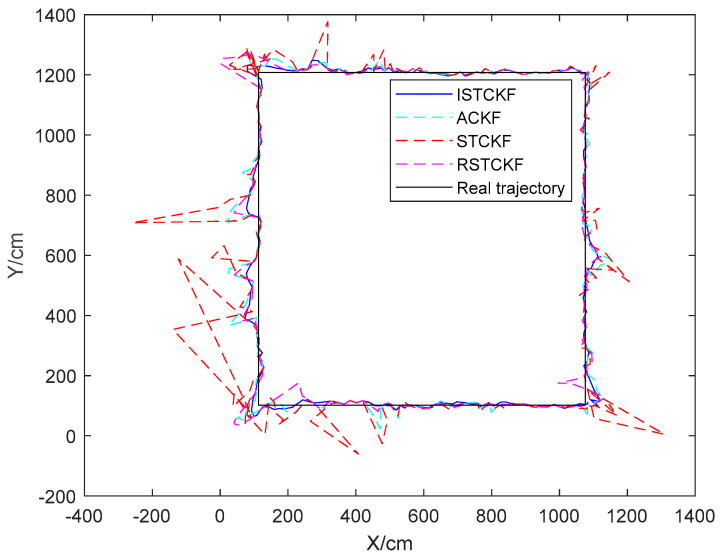
The tag positioning result of the algorithms.

**Figure 12 sensors-23-07463-f012:**
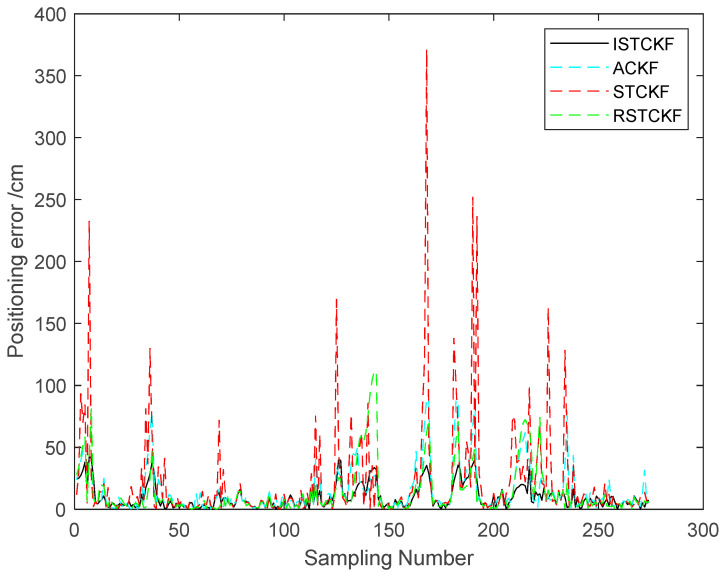
The positioning errors of the algorithms.

**Figure 13 sensors-23-07463-f013:**
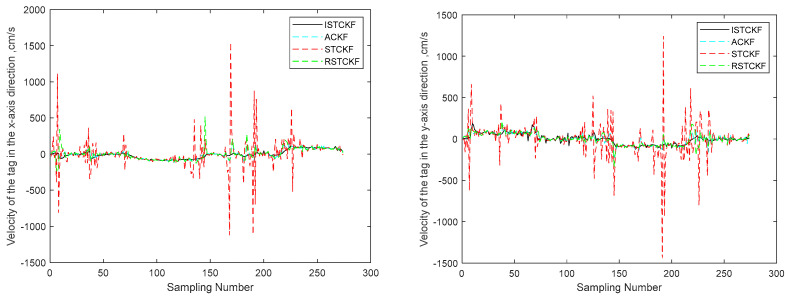
Estimated results of tag velocity of the algorithms.

**Table 1 sensors-23-07463-t001:** Improved strong tracking cubature Kalman Filtering algorithm.

Steps of the Proposed Algorithm
Step 1: $X_{k|k-1}$ and $P_{k|k-1}$ are obtained by the Equations (4)–(8) and $X_{k-1|k-1}$,$P_{k-1|k-1}$ and $Q_{k}$, where the Equation (4) is replaced by the Equations (65)–(67). Step 2: $MS_{i},i=1,2,\ldots ,\beta +1$ are calculated as inputs to the Equations (19)–(24) to obtain ${\hat{y}}_{1,k}$. Step 3: The Equations (25)–(36) are used to calculate ${\hat{R}}_{1,kj}$ and its threshold ${\hat{R}}_{1,kj,thr}$ and perform the primary NLOS identification. Step 4: Using the identification results obtained in step 3, perform the correction of the reconstructed observation ${\hat{y}}_{1,k}$ according to the Equations (37)–(45) to obtain ${\hat{y}}_{2,k}$. Step 5: Using ${\hat{y}}_{2,k}$ obtained in step 4 and the identification results obtained in step 3, perform the correction of the identification results according to the Equations (46)–(48). Step 6: Calculate the estimation ${\hat{R}}_{k}$ of the observation noise covariance matrix according to the Equations (49)–(52). Step 7: Using the identification results obtained from the step 5, $P_{k|k-1}$ is corrected according to the Equations (71)–(79) to obtain ${\hat{P}}_{k|k-1}$. Step 8: From the data obtained above, the final results $X_{k|k}$ and $P_{k|k}$ are calculated according to the Equations (9)–(18), where the Equation (9) is replaced by the Equations (68)–(70). Step 9: Using the previously obtained data, calculate $Q_{k+1}$. according to the Equations (53)–(64).

**Table 2 sensors-23-07463-t002:** Positioning error of each algorithm.

Positioning Algorithm		ACKF	STCKF	RSTCKF	The Proposed Algorithm
Average RMSE error in NLOS environment	Case 1 Case 2 Case 3 Case 4	8.40 cm 4.58 cm 12.46 cm 22.26 cm	27.41 cm 34.06 cm 93.57 cm 109.29 cm	14.86 cm 38.44 cm 76.00 cm 90.04 cm	1.13 cm 2.04 cm 4.05 cm 9.13 cm
Overall average RMSE error		2.1998 cm	4.6639 cm	3.4748 cm	0.9757 cm

Note: Cases 1, 2, 3, and 4 indicate the presence of 1, 2, 3, and 4 NLOS observations at the same moment, respectively.

## Data Availability

Not applicable.

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
