# Peer review of "Improved Strong Tracking Cubature Kalman Filter for UWB Positioning"

_sensors, 2023, doi:10.3390/s23177463_

Round 1

Reviewer 1 Report

This manuscript proposed an ISTCKF positioning algorithm for the problems of NLOS observation errors and inaccurate predictive dynamics model in UWB positioning system. The ISTCKF was compared to the ACKF, STCKF, and RSTCKF with simulation and field experiments. It can significantly improves the positioning accuracy.

(1)    What is the observation function h(·)?

(2)    In the description of the proposed positioning algorithm, the parameters’ physical meaning should be described. For example, is the observation vector yk distance between tag and anchor?

(3)    In Fig.8, why the reconstruction observations of BS1-4 are the same curve?

(4)     Check the Y-axis title of Fig.5 and 9. According to the state vector, position includes x and y, and how calculate the RMSE of position?

(5)    Is the proposed algorithm available for the long-term NLOS?

(6)    The state vector includes position and velocity. Can the results show some velocity information?

No comments.

Reviewer 2 Report

This manuscript proposes an improved strong tracking cubature kalman filter for UWB positioning. It conducted a simulation experiment and a field experiment in a small and ideal area. It seems that the experiments are well-written, but they lacks many key descriptions. Several issues should be solved.

1. The abstract lacks experimental results. Please modify it.

2. State estimation of nonlinear systems have two categories, linearization methods and sampling methods. EKF, UKF, CKF and PF belong to which category? 

3. Similar to the question 2, the identification methods on NLOS observations are classified into three categories, then references [31-33] belong to which category?

4. Please provide the references of ACKF, STCKF, and RSTCKF, or make them more explict.

5. Does the proposed IRACKF regard the mitigation of the positioning error of UWB positioning system as a unified mitigation of the model error and the NLOS errors? And provide detailed discussion in the discussion section.

6. Contributions should be rewritten, some of them are not contributions.

7. Why do both LOS noise and NLOS noise obey Gaussian distribution? What is the basis for such an assumption?

8. A flowchart of the proposed method should be added, as well as their simple description.

9. Tables in Section 4 lack unit, and several units are written in wrong format, such as 20×20 m, cm in italic.

10. Several evaluation metrics are widely utilized in the field of indoor positioning, e.g., mean absolute error, 75th percentile error, 95th percentile error, and RMSE. These metrics are well adopted in IPIN 2020 competition, PSOSVRPOS, A survey of recent indoor localization scenarios and methodologies. I strongly suggest that you could cite these references and add performance comparisons on these metrics.

11. In section 4.2, experiment description lack many key information,  such as sampling frequency, the height, the size, and the ground truth.

12. Figure 9 is very strange. Please check the lables of axes carefully.

13. English expression is poor, a polish is suggested.

English expression is poor, a polish is suggested.

Reviewer 3 Report

The topic of the submitted contribution is current and socially desirable. The authors of the paper proposed an Improved Strong Tracking Cubature Kalman Filter for UWB system. I have no fundamental comments about the mentioned UWB Receive Measurement Model and the performed experiment. I consider the simulation results to be correct. The simulation results show that the proposed algorithms make it possible to increase the accuracy of determining the position of the UWB system in the case of processing Gaussian processes. I consider the limitation of the modeling of random processes to Gaussian processes to be a lack of contribution. The formal administration of the contribution is at the required level.

Round 2

Reviewer 2 Report

All my concerns have been answered. Congratulations. This manuscript can be published.